# The long-term (24-month) effect on health and well-being of the Lifestyle Matters community-based intervention in people aged 65 years and over: a qualitative study

Robin Chatters,[1] Jennifer Roberts,[2] Gail Mountain,[1] Sarah Cook,[3] Gill Windle,[2] Claire Craig,[3] Kirsty Sprange[4]

[1]School of Health and Related Research, The University of Sheffield, Sheffield, UK
[2]Dementia Services Development Centre, Bangor University, Bangor, UK
[3]Health and Social Care Research Centre, Sheffield Hallam University, Sheffield, UK
[4]Faculty of Medicine & Health Sciences, Nottingham University, Nottingham, UK

**Correspondence to**
Robin Chatters;
r.chatters@sheffield.ac.uk

## ABSTRACT

**Objectives** To assess the long-term effect on health and well-being of the Lifestyle Matters programme.

**Design** Qualitative study of a subset of intervention arm participants who participated in the Lifestyle Matters randomised controlled trial (RCT).

**Setting** The intervention took place at community venues within two sites in the UK.

**Participants** A purposeful sample of 13 participants aged between 66 and 88 years from the intervention arm of the RCT were interviewed at 24 months post randomisation. Interviews aimed to understand how participants had used their time in the preceding 2 years and whether the intervention had any impact on their lifestyle choices, participation in meaningful activities and well-being.

**Intervention** Lifestyle Matters is a 4-month occupational therapy intervention, consisting of group and individual sessions, designed to enable community living older people to make positive lifestyle choices and participate in new or neglected activities through increasing self-efficacy.

**Results** Interviews revealed that the majority of interviewed participants were reportedly active at 24 months, with daily routines and lifestyles not changing significantly over time. All participants raised some form of benefit from attending Lifestyle Matters, including an improved perspective on life, trying new hobbies and meeting new friends. A number of intervention participants spoke of adapting to their changing circumstances, but there were significant and lasting benefits for 2 of 13 intervention participants interviewed.

**Conclusion** The majority of those who experienced the Lifestyle Matters intervention reported minor benefits and increases in self-efficacy, but they did not perceive that it significantly improved their health and well-being. The two participants who had experienced major benefits also reported having had life-changing events, suggesting that this intervention is most effective at the time when lifestyle has to be reconsidered if mental well-being is to be sustained.

**Trial registration** ISRCTN, ISRCTN67209155, post results.

**Strengths and limitations of this study**

► We present data obtained from a purposeful sample of 13 participants who received the Lifestyle Matters intervention within a randomised controlled trial.
► We analysed the interview data thoroughly using framework analysis.
► Recall may be an issue as participants were asked to recall events that occurred up to 2 years ago.

## BACKGROUND

Older adults can experience many life changing events in later life, such as retirement, disability and illness, which may have deleterious effects on mental and physical health and well-being.[1] In turn, decline and loss can result in social isolation and loneliness, which further erodes quality of life. Participation in meaningful activities can prevent this decline.[2 3] Occupational therapy aims to help individuals participate in both necessary and meaningful activities; it has an important role to play in health promotion and disease prevention by supporting engagement in activity and maintenance of occupational balance.[4]

The findings from recent systematic reviews support the use of psychosocial interventions for ageing well. Interventions identified through these reviews include engagement in meaningful social activities,[5] those designed to support occupational engagement[6] and multicomponent group-based health promotion interventions.[7] Guidance published by the UK National Institute for Health and Care Excellence (NICE) supports the use of group and individual sessions based on occupational therapy principles, where participants rather than facilitators are positioned as experts and peer support is fostered.[8]

Two studies undertaken in the USA (and recommended in NICE guidance) have investigated the clinical and cost-effectiveness of the Lifestyle Redesign intervention.[9 10] This is an occupational therapy intervention designed to support older adults at risk of decline in the participation in meaningful activities, thereby increasing their health and well-being.[9 10] Quantitatively, Lifestyle Redesign was found to improve the health and well-being of older adults living in a form of sheltered accommodation in a first study,[9 11] and in independently living older adults attending some form of community provision in a second study.[10] No qualitative work has been published to describe participants' experiences of the US intervention. Lifestyle Matters was inspired by the US Lifestyle Redesign and has been adapted to the UK context.

Based on an occupational approach to healthy ageing, the manualised Lifestyle Matters intervention is designed to assist participants to improve their well-being and avoid the decline associated with social isolation and poor mental health.[12] It involves participants meeting in a weekly group of up to 12 people over 4 months at a local community venue. Participants are also asked to engage in monthly individual sessions with one of the two facilitators at mutually agreed locations. Session topics can either be chosen from the manualised programme or new topics identified.[12] The facilitators work with the participants to explore the selected topic through discussion, activities and enactment in the community. The emphasis throughout is on the identification of participants' own goals, empowerment through sharing strengths and skills and providing support to enable them to practice new or neglected activities independently, particularly in the community. The intervention was postulated to change behaviour through Bandura's theory of self-efficacy, which theorises that self-efficacy is developed through four main sources—performance accomplishments (where success undertaking an activity builds self-belief in efficacy), vicarious experiences (seeing people similar to oneself undertake a task successfully), verbal persuasion that one has the ability to succeed in a given activity and inferences from emotional stimuli indicative of personal strengths and vulnerabilities.[13]

We undertook the Lifestyle Matters randomised controlled trial (RCT) to examine the clinical and cost-effectiveness of Lifestyle Matters, the protocol for which is reported elsewhere.[14] Between August 2012 and April 2013, 288 individuals living in an urban area in the North of England and 12 rural area in North Wales were recruited to the Lifestyle Matters study via general practitioner (GP) mail outs, community engagement and healthcare professional referrals. Individuals recruited to the study were randomised to either the intervention or usual care, in couples (if both individuals consented to participate in the study) or individually. Participants randomised to the intervention received the manualised 4-month Lifestyle Matters intervention, facilitated by NHS band four staff who were provided with preliminary training and supervised by qualified occupational therapists throughout

delivery. Both sets of participants were provided with questionnaires at baseline, 6 and 24 months. The primary outcome was the mental health component of the 36-Item Short Form Health Survey (SF-36) questionnaire, which was not significantly different between the control and intervention groups at either 6 or 24 months.[15 16] Participants were also interviewed qualitatively at 6 and 24 months to understand their experience of the intervention and any lasting effects.

Multi-methods research, where qualitative research is undertaken alongside quantitative methods, can be used in RCTs to explore the reasons for the quantitative findings and to explain variations in the effectiveness of an intervention.[17] Medical Research Council Framework recommends using qualitative research alongside trials 'to explain discrepancies between expected and observed outcomes, to understand how context influences outcomes'.[18] We therefore undertook a qualitative study 24 months post-randomisation to understand the long-term effects of the Lifestyle Matters intervention in order to ascertain if changes in well-being which were not identified quantitatively could be identified qualitatively.

## METHODS

The reporting of this study conforms to the Consolidated criteria for reporting qualitative research (COREQ) guidelines.[19]

### Recruitment and data collection

Interviews were undertaken at 24 months postrandomisation (approximately 18 months after intervention delivery had completed) by JR (Female Research Assistant with PhD) and RC (Male Trial Manager), both having experience of undertaking interviews with older adults. Participants were recruited to the study through purposeful sampling from the intervention arm of the study. All intervention participants who had been interviewed at 6 months and had provided 24-month quantitative follow-up data were approached to participate once more (n=13). Potential participants were first approached by JR or RC via a telephone call to ascertain initial interest in the study; no other previous contact was had between the researchers and participants prior to this. If interested, a participant information sheet and copy of the consent form was sent by post, and the participant was telephoned again to confirm that the information had been read, and he/she was happy to participate. Of those approached, nine participated in the substudy. Participants did not participate due to either not being contactable (n=2), not being interested in participating (n=1) or had already withdrawn consent from the main trial (n=1). Six additional intervention group participants were then approached to reflect a range of health-related experiences and abilities (based on their SF-36 scores at 24 months follow-up, age, location (North Wales/Northern England) number of intervention sessions attended and professional background (skilled, non-skilled) and

**Table 1** Demographics of interviewed participants

| Participant ID | Site | Age range | Gender | SF-36 MH score (6 months) | SF-36 MH score (24 months) | Number of group sessions attended | Number of one-to-one sessions received/ number offered |
|---|---|---|---|---|---|---|---|
| P14 | Northern England | 65–74 | Female | 60 | 70 | 13 | 1/4 |
| P15 | Northern England | 65–74 | Male | 85 | 65 | 12 | 1/4 |
| P16 | Northern England | 65–74 | Female | 65 | 80 | 14 | 2/4 |
| P17 | Northern England | 65–74 | Female | 90 | 80 | 9 | 0/4 |
| P18 | Northern England | 85–94 | Female | 90 | 70 | 12 | 1/4 |
| P25 | North Wales | 85–94 | Male | 95 | 95 | 13 | 2/3 |
| P20 | North Wales | 75–84 | Male | 70 | 90 | 15 | 1/4 |
| P21 | North Wales | 65–74 | Male | 70 | 90 | 15 | 1/4 |
| P22 | North Wales | 65–74 | Male | 90 | 90 | 11 | 1/4 |
| P23 | North Wales | 65–74 | Female | 90 | 95 | 14 | 3/4 |
| P24 | North Wales | 75–84 | Female | 85 | 100 | 1 | 0/0 |
| P19 | Northern England | 65–74 | Male | 90 | 70 | 15 | 1/4 |
| P26 | North Wales | 65–74 | Female | 85 | 80 | 15 | 1/4 |

SF-36, 36-Item Short Form Health Survey, scored on a 0 (poor) to 100 (good) health scale.

gender). Four of these consented, with participants not participating due to not being contactable (n=1) and not being interested in participating (n=1). In total, 13 participants were interviewed at 24 months. Participant demographics are summarised in table 1. The average age of participants was 73.0 years and ranged from 65 to 88 years. Six males were interviewed and six participants lived alone. The average SF-36 score of the sample was 82.7. One participant was in a couple with another study participant (who was not interviewed in this qualitative study).

Lifestyle Matters group attendance ranged from 1 to 15 sessions. Interviewed participants represented eight Lifestyle Matters groups, four in each site. The number of individual sessions offered to participants ranged between zero and four, the number actually received ranged between zero and three. Although they were not asked directly about it, no participants mentioned attending any one-to-one sessions in their interviews.

The semi-structured interview schedule aimed to explore, over the previous 18-month period (ie, from the end of the intervention phase): a) any changes in use of time and social participation; b) any impact that Lifestyle Matters on their lives and if so how and why.

The interview scheduled was piloted with a patient representative; the data from which was not used in the final analysis. Table 2 presents the questions which participants were asked, including example prompts.

All interviews were conducted in the homes of participants; no one else was present and repeat interviews were not undertaken. The participants would have been aware of the researcher's interest in the Lifestyle Matters intervention. Consent to audio record the interview was obtained from all participants and recordings were

**Table 2** Semi-structured interview questions and example prompts

| Question | Example prompt |
|---|---|
| How has life been for you over the last 18 months? | How has life changed for you over the last 18 months? Have things improved or got worse? |
| What does the term 'quality of life' mean to you? | What would improve the quality of your life? |
| How has life changed for you over the last 18 months? | Are you doing more or less activities (social or otherwise) |
| What do you think prevents you from engaging in social/leisure activities, or activities that you used to enjoy doing? | Has anything made it more difficult to do such activities? |
| Has the Lifestyle Matters group changed your life in any way? | Has the Lifestyle Matters group affected how easy or hard it is for you to engage in social activities? |

transcribed verbatim. Data collection was ended when the diversity identified in the sampling frame had been reflected in the interviews. Interviews lasted between 20 and 50 min and field notes were made after the interviews. Transcripts were not returned to participants for comments or corrections. Although sample size was prespecified, data saturation was discussed and confirmed following completion of all interviews; transcripts were read and themes identified to ensure that no new themes were arising in the final few interviews.

## Data analysis

A deductive framework approach was used for data analysis,[20] the starting point for which was the theory by which the intervention was postulated to function (as described above). Framework Analysis is a five-stage process of: familiarisation, forming a thematic framework, indexing, charting and mapping and interpretation. Familiarisation was achieved through reading and rereading a selection of interview transcripts. A thematic framework, consisting of major themes and one level of minor themes, was developed by two researchers (JR and RC). A small number of transcripts were coded using the framework, after which the researchers met to discuss the framework and finalise any changes that needed to be made. The same two transcripts were coded by three researchers (JR, RC and OO) to ensure comparable coding; any differences were discussed. All other transcripts were coded by one researcher using NVIVO V.10 software. Matrices were then developed, with one theme per matrix. Themes were derived from the data. Matrices were split into activities, attitudes and personal resources, health, mobility, quality of life (overview), social, family, community and other experiences and effects of Lifestyle Matters. These matrix charts were then examined for cross cutting themes and patterns in the data were mapped to inform the final level of interpretation. In order to report the results in a coherent manner, health, mobility and quality of life matrices were combined to form 'mental and physical health'. Attitudes and personal resources and community were combined to form 'personal and community resources'. New matrices were coded using elements of other charts to form the following new major themes: new skills and knowledge, change in routine, new leisure activities, meeting new people, family commitments and bereavement. We attempted to identify if the intervention increased self-efficacy through any of Bandura's four sources of self-efficacy (verbal persuasion, performance accomplishment, emotional arousal and vicarious experience), using the participant's own words. Participants did not feedback on the findings.

## Ethical considerations

All participants gave written informed consent for participation in the study. Participants were telephoned to ascertain their interest in participating, and were sent a consent form and information sheet that explained the purpose of the study. The consent form was then signed in the presence of a researcher (RC or JR) by the participant on the day of the interview, when consent to record the interview was also obtained. Ethical approval for the main trial and this qualitative substudy was granted by the South Yorkshire Research Ethics Committee (reference number 12/YH/0101). Transcripts were anonymised prior to data analysis in order to preserve confidentiality.

## RESULTS

Table 3 summarises the impact of the intervention on interviewed intervention group participants and the associated theoretical underpinning.

## Mental and physical health

Eleven participants reported having either a short-term or long-term health condition that had not changed significantly over the time period. Three participants discussed how their health had declined due to major illness; two had been receiving treatment for cancer and one participant's health had declined due to chronic obstructive pulmonary disease. The intervention did not reportedly have any impact on the physical health of participants in the intervention group.

The intervention reportedly had a positive effect on mental well-being. Four participants stated that attending the intervention helped them to 'gain perspective' on their lives and what they have. For two of these, the change in perspective helped to broaden their outlook.

> …it's … helped me with … knowing what you could do… if there's a group of us, probably someone comes up with a suggestion or something you know and … it's probably broadened my outlook a little bit more. (P18, Female, Northern England)

> It has made me look at things in a different way. As I said, there are answers to be had, you don't have to give up, there is a way forward and there are people willing to help you too. (P22, Male, North Wales)

The other two participants related the apparent social isolation of fellow participants in the group to their own situation, helping them to appreciate what they have.

> (Lifestyle Matters has) given me an idea how the other half live, in a way, and I suppose in a way I was very pleased with my … position in life. (P25, Male, North Wales)

> …it makes me appreciate … my friends and my husband still being with me and the life that I have and my family that I have got… (P24, Female, North Wales)

It should be noted that this latter participant (P24) only attended one intervention session, so it is unlikely that the intervention had an effect on self-efficacy.

Other participants discussed how being involved had improved their mental well-being. Two described how attending the group aspect of the intervention provided

**Table 3** Impact of the Lifestyle Matters intervention and associated theoretical underpinning

| Participant ID | Benefit(s) | Self-efficacy source |
|---|---|---|
| P14 | Started computer lessons and now uses it regularly | Performance accomplishment |
| P15 | Took up Bridge following support from group | Verbal persuasion |
| P16 | Confidence and support after death of husband | Performance accomplishment |
| P17 | Made new friends<br>Joined a keep fit class following support from group | Performance accomplishment |
| P18 | Broadened outlook | Emotional arousal |
| P25 | Gave perspective seeing how 'less fortunate' people live | Emotional arousal |
| P20 | Improved confidence to socialise with new people; now socialises more | Performance accomplishment/vicarious experience |
| P21 | Helped 'pull out of misery' following illness | Emotional arousal |
| P22 | Gained perspective<br>Helped him 'find answers' (now using credit union) | Emotional arousal |
| P23 | Improved confidence in group situations | Performance accomplishment/vicarious experience |
| P24 | Gained perspective | Emotional arousal |
| P19 | Increased confidence to undertake new activities<br>Mental lift<br>Positive impact on spousal relationship | Performance accomplishment<br>Emotional arousal |
| P26 | Undertaking rock climbing with group has provided her with increased confidence to do other things<br>Provided healthy eating information | Performance accomplishment |

them with a 'mental lift', which lasted a short period of time after each session.

> …it give us a buzz which gave us a mental lift which lasted for 2 or 3 days after…it's when you talk about it that you think yes perhaps I should be joining some activities and I would get that again perhaps. (P19, Male, Northern England)

This had positive effect on the participant's relationship with his wife.

> because of what people said at the group made me appreciate how good she is and made me feel quite content, I have a lot to be happy, a lot to be happy and contented about but I probably didn't realise. (P19, Male, Northern England)

Another participant spoke of how the intervention helped to pull him out of misery following illness.

> Yes I was still having to go for outpatients appointments… but when you are sort of with a group and … it's a specific day you are motivated and I think … talking with the group and laughing and joking sort of spurs you on…you know pulls you out of your misery like. (P21, Male, North Wales).

The improvement of these participant's mental state, and the 'gaining of perspective' of other participants may have impacted on their somatic and emotional status, allowing participants to interpret stress reactions more positively and therefore possibly increasing their self-efficacy.

### New skills and knowledge

Around half of those interviewed who had received the intervention stated that they had gained new skills and knowledge from attending Lifestyle Matters. This included information about financial matters, healthy eating and services that can help older people. The majority of participants described gaining knowledge about financial matters and local adult education classes, including computers and keep fit.

> we talked quite a lot at …about eating and just sort of how we lived generally and I think it did just sort of widen our scopes a little bit… (P26, Female, North Wales)

### Adaptation to leisure activities

Even though, for the majority of participants, their health did not decline over the time period, physical health affected the activities participants were able to undertake. Participants had to stop or reduce certain activities due to their health. One participant used to walk regularly, but due to physical health/mobility limitations has instead started a new hobby (cooking), although this was not instigated by the Lifestyle Matters intervention.

> I use to enjoy going to the country and going for a walk, I've had to cut down on the (walking) that would help me…. [I've cut down because of] my knee, my back, my physical health. One of my main

hobbies is cooking and that gives me pleasure. (P22, Male, North Wales)

Another stopped knitting due to dexterity and eyesight problems.

I used to do a lot of knitting I used to knit for the family and I used to sew my own clothes, but I don't do that anymore because I haven't got the same dexterity as I used to have and my eyesight is not as good as it was. (P24, Female, North Wales)

Learning about new activities as part of Lifestyle Matters meant that some participants developed new leisure pursuits, including computing, keep fit and attending opera with a group.

Three participants had discovered a new hobby through participation in the intervention and were still continuing the hobby at the time of their interview. One of these participants met new friends through the intervention and joined a keep fit class. Making a new friend appeared to provide a performance accomplishment for this participant.

You meet one person and then through that person you meet their friends… then you… join up with them and make friends then you start doing things with them. (P17, Female, Northern England)

Eight participants did not describe getting involved in new activities. They said that they already led busy lives and therefore did not have the time or impetus to continue with new activities.

Two participants had pursued and enjoyed new activities during group sessions, but did not continue these after the intervention ended. One of which tried and enjoyed crazy golf but did not continue with it; trying the new activity provided him with confidence to try other new activities in the future.

Some of the activities that we took part in be it crazy golf, visiting museums I did intend to go back and have a go at some of these activities but I have slipped back into my old… my own comfortable lifestyle. (P19, Male, Northern England)

Attending the group was good in [the] respect that I know there is things out there and if I want to go and have a try I can go and do it. (P19, Male, Northern England)

The second mentioned enjoying climbing during the intervention. She had not continued with this after the intervention had finished, but doing so had increased her confidence; as a result, she had the confidence to undertake an activity that she never thought were possible.

It's certainly given me the inclination to do the zip wire. Because we did the … wall climbing. I felt that doing that was, wow, it was just something that I never thought that I would be able to do. And the fact that I practically reached the top of the wall, I was only one grip away….and to be able to get up there and get down again, I thought was just amazing… I never thought that I would do that and I did it, means that I can do the zip wire if I feel so inclined. (P26, Female, North Wales)

The intervention provided both participants with a 'performance accomplishment', allowing them to succeed in a new hobby or activity in a safe environment. This positive experience built up their belief in their self-efficacy, making it much more likely that they would try the new activity (or other new activities) again. This was also true for other participants who tried computing and attending opera with the group.

One participant described how participation in the intervention had a major impact on his life. He had been a keen golfer but his health had deteriorated and he had had to give up golf a year before the study commenced. He had continued to attend the social events at the golf club for a period of time but then this had also stopped because he no longer had as much in common with the other members. He reported how he had started to become lonely, and Lifestyle Matters helped him to discover a new interest.

Bridge seemed like a good idea, but I'd never got off my backside and done anything about it, and the two girls that took the course were very good at kicking me up the backside and making sure I did do something about it, for which I've been very grateful as that's made a huge difference to my life. (P15, Male, Northern England)

The intervention facilitators provided this participant with verbal persuasion in order to help him increase his confidence to initiate a new activity.

### Meeting new people

All participants met new people as part of the Lifestyle Matters groups, but only a minority made 'meaningful' relationships that lasted past the end of the group. Around half of the interviewed intervention participants stated that they had not continued meeting with the group as they had not met people with similar interests.

…although the people were all right they were nice people there was nobody really that I would have wanted to have kept on with you know (P14, Female, Northern England)

Six participants had plans to meet up with their fellow group members after the intervention had finished, but this only lasted for a short period of time, as the person who volunteered to do the organising had not been in touch.

One of the members had promised to contact each one of us, he had all our telephone numbers, but he didn't do anything. (P17, Male, North Wales)

Further group meetings also stopped after a few weeks or months because people started to drop out.

We met up for, I think we were going to meet up one time and then everybody started going on holiday so it weren't possible but and I think it has just filtered off as I haven't seen them. (P16, Female, Northern England)

One participant was still meeting on a one-to-one basis with a fellow member of the Lifestyle Matters group, either talking by telephone or undertaking activities together, and subsequently making more friends.

just by going to that health thing that we went to up there through this, that [Lifestyle Matters] group, I made some friends there that we still go out with, a couple of ladies and then a walking group we joined. (P17, Female, Northern England)

Meeting new people in a group situation provided two participants with self-efficacy via either performance accomplishment and/or vicarious experience; confidence was gained to interact with people in such situations.

Participant: "I've got more confidence now in myself then I used to have years ago yeah".
Interviewer: "…do you think that's anything to do with the groups…"
Participant: "It just makes me talk a bit more perhaps". (P23, Female, North Wales)

It might have been [the Lifestyle Matters group] … might have given me more confidence to walk into a room full of women… (P20, Male, North Wales)

## Family commitments

Family commitments influenced participant's use of time, with three interviewees reporting that family was a major part of their routine. One of those participants reported an increased commitment to taking care of family because he had recently started taking care of his grandchildren full time, which had resulted in a change in use of time:

Apart from looking after the little girls full time, that's changed my life in the past eighteen months or so. This has restricted my personal activities a lot. So I sometimes feel that I don't get enough time to do what I want to do. (P17, Male, North Wales)

A frequently described situation was that of having to undertake routines to fit in with the work-life routines of family, specifically in order to take care of grandchildren. One participant commented that she did not feel the balance was currently in favour of her personal interests:

because family when they are working and that you seem to get set days and that can you know so you end up doing that (….) whether you want to or not yeah you know they rely on you so you feel obliged to do it….At the moment no because I have had some of my

grandchildren poorly so that has gone a bit haywire at the moment. (P16, Female, Northern England)

None of those interviewed described any impact on how they negotiated family commitments that could be attributed to the intervention.

## Personal and community resources

Availability of community resources reportedly affected the ability of some participants to undertake activities outside the home; four participants stated that they had been unable to continue to attend classes or the gym because of various reasons such as gym closure, a class having too few members to be viable any longer and death of the person who ran a class.

I have not been able to do as much Thai Chi because I used to be able to do it 3 times a week and then because [instructor] didn't get many people going on a Wednesday he stopped doing it. (P14, Female, Northern England).

The participants who mentioned these challenges were already either attending several other classes, or were able to find a new gym, highlighting their desire to remain proactive. Lifestyle Matters aided them to locate new activities, with the participant who had to stop Thai Chi taking up computer classes instead; the facilitators assisted her to experience performance accomplishment, which resulted in her attending the class again

[The facilitator] took us down [to the computer class] and introduced us…we have been ever since and we go every week. (P14, Female, Northern England).

## Bereavement

Bereavement was mentioned by a number of participants. Death of a spouse, family member or friend had a marked effect on lifestyle.

All participants who had lost a partner spoke of having to adapt and develop in some way, including developing new skills or re invigorating neglected ones and engaging in new social activities.

…when you make new friends when you lose the friends that you have got if their still with their husbands it is, I don't know it makes you feel a bit, its hard to put into words sometimes [pauses] [cries] it is an emptiness and you have got to try and fill this emptiness yes [pause] the only way to do it is to, this is me, is to make myself do things and go out and speak to people. (P16, Female, Northern England)

Engagement in the intervention reportedly had a major impact on this participant's life. This woman's husband had died prior to her involvement in Lifestyle Matters. Participation reportedly helped her to deal with the recent bereavement which had left her low in confidence and with fewer social opportunities.

when you end up on your own you don't go out of these four walls I would think you can lose that confidence and very quickly. (P16, Female, Northern England)

Attending Lifestyle Matters helped her to regain her confidence and re-engage in social activities.

I think it gives you something some kind of confidence when you when you know you do end up on your own and err and I just think it gives you confidence because to meet people and to be in a group.(P16, Female, Northern England)

It is clear that the intervention provided this participant with a 'performance accomplishment'—she was nervous about going alone to a new group of strangers, but having succeeded to make friends, she has more confidence to undertake such an activity again.

## DISCUSSION

This qualitative study found that those who had participated in Lifestyle Matters reportedly enjoyed the intervention and all spoke of receiving some form of benefit but only 2 out of the 13 we interviewed were able to describe any major long-term impact. Many participants described a process of adapting the type of activity they undertook following changes in their circumstance—two participants swapped active hobbies for sedentary ones (e.g., walking for cooking) following mental or physical decline; other participants, following a change in community resources, had to find new activities. A few participants had to adapt their activities following bereavement of their partner. Participants displayed self-efficacy to adapt to their new circumstances, although only two of the interviewed intervention participants were able to attribute this adaptability to the intervention. It is significant therefore that these two individuals had recently experienced a challenge in their lives immediately prior to their engagement in the intervention; one had been recently bereaved and the other had recently had to adjust to a reduction in mobility. This finding indicates that the Lifestyle Matters intervention was most meaningful for those at a 'trigger point' in their life which demanded transition and change. The intervention assisted those in the intervention arm who were at this trigger point to adapt to their new circumstances. Lifestyle Matters provided sign-posting to new activities for those individuals whose current activities were no longer available. Self-efficacy was boosted by providing participants with performance accomplishment (where participants succeeded at new activities) and modelling influences (where participants witness their peers undertake activities successfully).[13] Emotional arousal was provided by improving mental health and participant's outlook on life. It is clear that for two participants, the intervention assisted an increase in self-efficacy, which resulted in change in behaviour and increases in quality of life. For other intervention participants, there are examples described above of their self-efficacy increasing, but this did not result in a major change in their day-to-day activities and resulting quality of life. It is evident that barriers were present which prevented participants from adapting—barriers included family commitments, physical and mental health, community resources and bereavement.

This study has a number of strengths. Interviews were undertaken and analysed by two researchers, ensuring that objectivity was maximised. It should be noted that these interviews were undertaken during the winter at a time of prolonged snow and icy conditions; such weather conditions have previously been shown to reduce physical activity levels in older adults with self-reported limitations in activities of daily living.[21] Participants were asked to discuss life events covering a 2-year period. Recall may have also been an issue, especially when discussing the impact of an intervention they received 2 years previously. Participants experience of the intervention and resulting increases in self-efficacy were linked to Bandura's four sources of self-efficacy through analysing participant's experience of the intervention and associated benefits. It was difficult to determine if possible increases in self-efficacy (e.g., increased confidence) were due to vicarious experience or mastery experiences, as participants did not discuss explicitly which aspect of the intervention led to such changes.

This qualitative study highlights that in the Lifestyle Matters study, only a small number of participants were at a 'trigger point'. This is in contrast to the US Lifestyle Redesign studies, and the UK feasibility study which was undertaken prior to the RCT, which suggest that all participants gained from the intervention in a meaningful way when assessed in the USA[10 22] and in the UK.[1] However, this difference may be explained by the lack of long-term qualitative follow-up in these preceding studies; qualitative work was not undertaken in the US studies, and the UK feasibility study qualitatively followed up participants directly after the cessation of the facilitated group sessions and therefore did not assess the long-term impact of the intervention.

The quantitative and qualitative findings from this work did not demonstrate that the intervention had any impact on quality of life at population level. The majority of those recruited to participate reported being busy, active and resourceful, engaged in a diverse range of activities and with established social networks and it can therefore be argued that they had little to gain from the intervention. Inability to identify those at risk of mental decline appears to be a prime reason for the findings obtained through both quantitative and qualitative enquiry. Another issue that may have contributed includes the duration of the intervention (4 months), which was far shorter than for the US Lifestyle Redesign studies (6 months duration) and for the Lifestyle Matters feasibility study (8 months duration).[10 22] A longer delivery duration can give those with potentially 'hidden' challenges the time to become sufficiently confident to work with others to explore

solutions. In addition, it can be argued that implicit issues such as social desirability and unacceptable disease bias,[23][24] which have been identified in other qualitative studies,[25] may mean that the benefit of the intervention was under-reported. One of the aims of the group sessions was to aid participants in moving forward together with all parties taking an active role in organising outings and meet ups. Unfortunately, the majority of participants did not continue in their groups after the cessation of the intervention, often because the person who volunteered to continue to organise the groups did not make contact with the other group members.

Clinicians and policy makers should consider who is best to facilitate an intervention such as Lifestyle Matters. In the current study, the facilitators were NHS band four therapists, who did not have in-depth training in how to manage such groups (although they did receive study-specific training). More highly trained facilitators, such as occupational therapists, may have improved the delivery of the intervention. In addition, the Lifestyle Matters intervention could be further developed in order to improve the benefit to participants. Although they were not asked specifically, none of the interviewed participants mentioned the one-to-one sessions in their interview or stated they had gained any benefit from them; the intervention could therefore be further developed by increasing the understanding of the purpose of the one-to-one sessions with both the intervention participants and facilitators.

Future research needs to be undertaken in order to identify how isolated older adults can be identified and recruited to preventative health interventions such as Lifestyle Matters. The fact that only 2 out of 13 intervention participants gained 'major' benefit from the intervention could be due to the recruitment methods used in this study, where participants were recruited, in the majority, via GP mail outs, resulting in a physically and mentally well population. The majority of participants who were interviewed in this study were not at a 'trigger point' in their life when they were in need of a health and well-being intervention such as Lifestyle Matters. Despite the emphasis current guidance places on recruiting isolated older adults to preventative lifestyle interventions, other studies have also discussed issues with identifying such individuals.[8][26][27] Future evaluations should be undertaken to assess the viability of other recruitment techniques, such as respondent-driven sampling, which has been used previously to recruit isolated adults with HIV.[28]

**Contributors** RC, KS, JR, SC, CC, GM and GW designed the study. RC and JR undertook data collection. RC, JR and SC analysed the data. RC and JR wrote the first draft of the manuscript; all authors reviewed and approved the manuscript.

**Funding** This work was supported by the Medical Research Council grant number (G1001406).

**Competing interests** None declared.

**Ethics approval** Ethical approval for the main trial and this qualitative substudy was granted by the South Yorkshire Research Ethics Committee (reference number 12/YH/0101). All participants consented to participate in this qualitative study.

**Provenance and peer review** Not commissioned; externally peer reviewed.

**Data sharing statement** All relevant data are published within this manuscript.

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
