## [Reviewer comments · BMJ Open]

ARTICLE DETAILS

TITLE (PROVISIONAL)	The long-term (24 month) effect on health and wellbeing of the Lifestyle Matters community based intervention in people aged 65 years and over: a qualitative study
AUTHORS	Chatters, Robin; Roberts, Jennifer; Mountain, Gail; Cook, Sarah; Windle, Gill; Craig, Claire; Sprange, Kirsty

VERSION 1 - REVIEW

REVIEWER	Tamlin S Conner University of Otago, New Zealand
REVIEW RETURNED	17-Apr-2017

GENERAL COMMENTS	This article presents the qualitative results of interviews with older adults who took part in the Lifestyle Matters intervention (or control). The interviews took place 24 months post randomisation; my understanding is the intervention was a 6 month intervention designed to improve the health and well-being of older adults. I offer the following comments: Major Concerns 1. There needs to be a better rationale for conducting this study. Why is it important to gain knowledge from this intervention study which otherwise failed to show any significant improvement in well-being between the intervention and control groups? What knowledge is there to gain from interviewing people in an intervention shown NOT to improve outcomes? There needs to be better justification in the introduction for doing this research. For example, one objective could be to understand whether differences in well-being not observed quantitatively could be observed qualitatively? Or, an objective might be to understand why they think the intervention didn't have lasting benefits. What knowledge is there to gain from a failed intervention? I felt that the discussion touched on some important knowledge gain (such as finding that the intervention might be best suited to people at a "trigger point" - as a means of providing support during difficult transitions in older adulthood).2. The 18-month recall period is a major concern. This is quite a lot of time to ask a person to recall their experiences about the intervention. However, if the aims were refined to focus on long term change - then an examination this long after the intervention could make sense. Minor Concerns
--

	1. Have the null quantitative results from this intervention been published? This should go where reference [16] is currently placed. 2. For Table 1, explain the scale of the SF-36 MH score (i.e., that it ranges from 0 to 100 in 20 point increments with higher values indicating greater mental health). This will help readers to see that the mental health scores were fairly high. 3. Page 10: "one-one" should be "one-to-one" to be consistent with latter language. 4. Please provide detail on questions asked in the semi-structured interview. There needs to be more detail on the types of prompts that were used by interviewers. 5. For the semi-structured interview aim a): what time period were they asked to describe changes in time usage and social participation. During the intervention only? From then until now? Please provide timeframe. 6. Clarify whether intervention facilitators were NHS band four or band three staff. Page 5 describes them as band four. Page 24 describes them as band three. 7. Page 22 - It is written that that the "interviewer were undertaken and analysed by two researchers, ensuring that objectivity was minimised?" Did you mean so that objectivity was maximised? 8. I was confused by the interpretation that the timeframe of delivery of the intervention could have been the cause for the null results. It is pointed out that the US Lifestyle intervention was 6 months long. Wasn't this intervention also 6 months long?
--	---

REVIEWER	Holly Gwyther Aston University United Kingdom
REVIEW RETURNED	08-May-2017

GENERAL COMMENTS	While this is an interesting paper which covers an interesting and clinically relevant topic, I have some concerns which need addressing in order to improve the paper. 1. The main conclusion that the lasting benefits of the intervention can be ascribed to receptivity to change following major life events, is an important one and could be developed further with reference to health psychology models including the Theory of Planned Behaviour (Ajzen, 1991) and Health Belief Model. For example, the "trigger points" could be acting as a "cue to action". Further, within their theme of 'New Leisure Activities', the authors describe a participant's change in confidence levels in undertaking new activities which could also be ascribed to changes in Perceived Behavioural Control (PBC) and self-efficacy. Similarly with P16 under the 'Bereavement' section. Furthermore, in Table 2: Impact of the Lifestyle Matters Intervention, the benefit(s) column could really read as a mechanism of action, either as a change in attitude, change in PBC or change in behaviour. I think that appraising the information in light of these theories would add significantly to the work. Methods: 2. While the results have been divided up into short and coherent sections, I cannot see the links between the deductive framework described in the methods section and the final themes. 3. The use and reporting of the COREQ framework guidelines was excellent. 4. As you are comparing groups, it would be useful to have an overview of the demographics of each group, e.g., mean age, age
---

	range, gender split, mean SF-36 score. Results: 5. When reading the themes, they appear to fall broadly into two categories – benefits of the intervention, e.g., opportunity to meet new people, socialise and try new things; and barriers to success, e.g., deterioration in physical health, family commitments, resources, bereavement and weather. While the former are interesting in terms of the intervention success per se, it is how people overcome the barriers and make lasting change to maintain their connection to the social world that is really of interest here. There are examples of participants who adapted to physical incapacity by switching hobbies, e.g., from golf to bridge, and from walking to cooking. This demonstrates resilience and adaptability which are important characteristics in success in older age. This could be developed in the paper. 6. There are examples of themes which add little of interest to the paper, e.g. 'Routine'. 7. Although the results are presented as a comparison between the intervention group and the control group, I'm not sure that this is necessary, or adds much to the paper. Occasionally views of the control group, e.g. on mental wellbeing, were not discussed/explored. Unfortunately in its present form, the conclusions are not surprising and do not move the field forward substantially. However, the intervention and findings are of interest and could perhaps be more usefully explored by grounding within a theoretical health psychology framework.
--	--

VERSION 1 – AUTHOR RESPONSE

REVIEWER 1

1) There needs to be a better rationale for conducting this study. Why is it important to gain knowledge from this intervention study which otherwise failed to show any significant improvement in well-being between the intervention and control groups? What knowledge is there to gain from interviewing people in an intervention shown NOT to improve outcomes? There needs to be better justification in the introduction for doing this research. For example, one objective could be to understand whether differences in well-being not observed quantitatively could be observed qualitatively? Or, an objective might be to understand why they think the intervention didn't have lasting benefits. What knowledge is there to gain from a failed intervention? I felt that the discussion touched on some important knowledge gain (such as finding that the intervention might be best suited to people at a "trigger point" - as a means of providing support during difficult transitions in older adulthood).

Response: An improved rationale has been added to the background

2) The 18-month recall period is a major concern. This is quite a lot of time to ask a person to recall their experiences about the intervention. However, if the aims were refined to focus on long term change - then an examination this long after the intervention could make sense.

Response: The aim of the paper is stated on page 6 to "gain perspectives on lifestyle over the preceding 24 months, and identify whether the Lifestyle Matters intervention had any impact upon the lifestyle of those who experienced it compared to those who did not." Therefore, the aim is to assess long term change. This has been clarified in the background.

3) Have the null quantitative results from this intervention been published? This should go where

reference [16] is currently placed.

Response: This reference has been added.

4) For Table 1, explain the scale of the SF-36 MH score (i.e., that it ranges from 0 to 100 in 20 point increments with higher values indicating greater mental health). This will help readers to see that the mental health scores were fairly high.

Response: This has been added to table 1.

5) Page 10: "one-one" should be "one-to-one" to be consistent with latter language.

Response: This has been altered.

6) Please provide detail on questions asked in the semi-structured interview. There needs to be more detail on the types of prompts that were used by interviewers.

Response: This has been added to the manuscript (Table 2).

7) For the semi-structured interview aim a): what time period were they asked to describe changes in time usage and social participation. During the intervention only? From then until now? Please provide timeframe.

Response: This has been added to page 11.

8) Clarify whether intervention facilitators were NHS band four or band three staff. Page 5 describes them as band four. Page 24 describes them as band three.

Response: This has been corrected.

9) Page 22 - It is written that that the "interviewer were undertaken and analysed by two researchers, ensuring that objectivity was minimised?" Did you mean so that objectivity was maximised?

Response: This has been corrected

10) I was confused by the interpretation that the timeframe of delivery of the intervention could have been the cause for the null results. It is pointed out that the US Lifestyle intervention was 6 months long. Wasn't this intervention also 6 months long?

Response: The UK Lifestyle Matters intervention was 4 weeks in duration (as stated on page 5).

Clarification has been added to the discussion regarding this (page 23).

REVIEWER 2

1) The main conclusion that the lasting benefits of the intervention can be ascribed to receptivity to change following major life events, is an important one and could be developed further with reference to health psychology models including the Theory of Planned Behaviour (Ajzen, 1991) and Health Belief Model. For example, the "trigger points" could be acting as a "cue to action". Further, within their theme of 'New Leisure Activities', the authors describe a participant's change in confidence levels in undertaking new activities which could also be ascribed to changes in Perceived Behavioural Control (PBC) and self-efficacy. Similarly with P16 under the 'Bereavement' section. Furthermore, in Table 2: Impact of the Lifestyle Matters Intervention, the benefit(s) column could really read as a mechanism of action, either as a change in attitude, change in PBC or change in behaviour. I think that appraising the information in light of these theories would add significantly to the work.

Response: A theoretical basis has been added to the manuscript.

The theory is postulated to have an effect via Bandura's theory of self efficacy. Detail about this has been added to the introduction, results (including table 2) and the discussion.

2) While the results have been divided up into short and coherent sections, I cannot see the links between the deductive framework described in the methods section and the final themes.

Response: Extra detail has been added to the methods to describe how the deductive framework was

used to produce the final themes.

3) As you are comparing groups, it would be useful to have an overview of the demographics of each group, e.g., mean age, age range, gender split, mean SF-36 score.

Response: Although the paper no longer compares groups, this has been added to page 7 of the manuscript.

4) When reading the themes, they appear to fall broadly into two categories – benefits of the intervention, e.g., opportunity to meet new people, socialise and try new things; and barriers to success, e.g., deterioration in physical health, family commitments, resources, bereavement and weather. While the former are interesting in terms of the intervention success per se, it is how people overcome the barriers and make lasting change to maintain their connection to the social world that is really of interest here. There are examples of participants who adapted to physical incapacity by switching hobbies, e.g., from golf to bridge, and from walking to cooking. This demonstrates resilience and adaptability which are important characteristics in success in older age. This could be developed in the paper.

Response: Adaptability has been developed within the paper and is now discussed in the discussion.

5) There are examples of themes which add little of interest to the paper, e.g. 'Routine'.#

Response: This section has been removed.

6) Although the results are presented as a comparison between the intervention group and the control group, I'm not sure that this is necessary, or adds much to the paper. Occasionally views of the control group, e.g. on mental wellbeing, were not discussed/explored.

Response: We agree. The control group does not add much to the paper and has been removed from the analysis.

7) Unfortunately in its present form, the conclusions are not surprising and do not move the field forward substantially. However, the intervention and findings are of interest and could perhaps be more usefully explored by grounding within a theoretical health psychology framework.

Response: As discussed above, such theoretical grounding (i.e. Bandura's theory of self efficacy) has been added.

VERSION 2 – REVIEW

REVIEWER	Holly Gwyther Aston Research Centre for Healthy Ageing Aston University Birmingham B4 7ET
REVIEW RETURNED	16-Jun-2017

GENERAL COMMENTS	1. I was pleased to note that the authors have added a theoretical basis to the manuscript. The authors postulate that the intervention has an effect via Bandura's theory of self-efficacy. I was particularly interested in Table 3 where the source of the change has been attributed. It is my understanding that the authors have allocated the source of self-efficacy based on participants' own words. Certainly P26 suggested that the feeling of mastery over the rock climbing had given her the confidence and self-efficacy to attempt other challenges which appeared to lead to the label 'performance accomplishment'. If so, this could be better described in the
---

	methods. 2. For a number of participants, the self-efficacy source was not obvious either from the quotations used or the explanatory text. This needs clarifying - perhaps phrases could be identified from the transcripts and added to the table, or clarified in the thematic discussion somewhere? 3. I was surprised that no explicit evidence was found for vicarious experiences. However, it may be that these were indirect influences, certainly where improved confidence is mentioned as a benefit in the table. 4. I note that a claim is made for the intervention in terms of a positive effect on mental wellbeing. However, P24 quoted here had only taken part in one of a potential 15 sessions (unless this is a typo). As a consequence, I'm not sure that the evidence is here yet to support this. Certainly, the evidence does suggest that the intervention improved participants' immediate affective state, giving them an emotional 'lift' and an opportunity to reflect on their own lives and experiences, particularly in comparison with others. 5. Given the inclusion of the theoretical basis within the thematic description, the themes could do with restructuring and/or reordering. 6. The discussion reads very well and makes a number of interesting and valid points, worthy of publication.
--	---

VERSION 2 – AUTHOR RESPONSE

Reviewer: 2

Holly Gwyther

Aston Research Centre for Healthy Ageing, Aston University, Birmingham. B4 7ET

Please state any competing interests or state 'None declared': None declared

Please leave your comments for the authors below

Thank you for the opportunity to read the amended draft of this paper.

1. I was pleased to note that the authors have added a theoretical basis to the manuscript. The authors postulate that the intervention has an effect via Bandura's theory of self-efficacy. I was particularly interested in Table 3 where the source of the change has been attributed. It is my understanding that the authors have allocated the source of self-efficacy based on participants' own words.

Certainly P26 suggested that the feeling of mastery over the rock climbing had given her the confidence and self-efficacy to attempt other challenges which appeared to lead to the label 'performance accomplishment'. If so, this could be better described in the methods.

RESPONSE: Detail has been added to the methods to describe that sources of self-efficacy were identified from participant's own words.

2. For a number of participants, the self-efficacy source was not obvious either from the quotations used or the explanatory text. This needs clarifying - perhaps phrases could be identified from the transcripts and added to the table, or clarified in the thematic discussion somewhere?

RESPONSE: For all participants, the source of self-efficacy is now clarified in the thematic discussion.

3. I was surprised that no explicit evidence was found for vicarious experiences. However, it may be

that these were indirect influences, certainly where improved confidence is mentioned as a benefit in the table.

RESPONSE: We have added vicarious experiences to the table of self-efficacy sources; as the reviewer points out, we cannot be sure of the exact source of self efficacy for some intervention benefits (especially increases in confidence), because the participants did not directly state how (or if) self-efficacy was increased. This has been reflected in the limitations of the study.

4. I note that a claim is made for the intervention in terms of a positive effect on mental wellbeing. However, P24 quoted here had only taken part in one of a potential 15 sessions (unless this is a typo). As a consequence, I'm not sure that the evidence is here yet to support this. Certainly, the evidence does suggest that the intervention improved participants' immediate affective state, giving them an emotional 'lift' and an opportunity to reflect on their own lives and experiences, particularly in comparison with others.

RESPONSE: This is now reflected within the results.

5. Given the inclusion of the theoretical basis within the thematic description, the themes could do with restructuring and/or reordering.

REPOSENSE: I am unsure how the themes can be restructured to improve the results section. The results are currently structured and ordered by over-arching theme, and theoretical basis is included where it is clearly evident from the data that a source of self-efficacy has been affected. Any other way of structuring the results may confuse the reader.

6. The discussion reads very well and makes a number of interesting and valid points, worthy of publication.